# Preparation of Methacrylate-Based Polymers Modified with Chiral Resorcinarenes and Their Evaluation as Sorbents in Norepinephrine Microextraction

**DOI:** 10.3390/polym11091428

**Published:** 2019-08-30

**Authors:** Alver Castillo-Aguirre, Mauricio Maldonado

**Affiliations:** Departamento de Química, Facultad de Ciencias, Universidad Nacional de Colombia-Sede Bogotá, 30 No. 45-03, 7122 Carrera, Colombia

**Keywords:** Chiral resorcinarene, aminomethylation, microextraction, norepinephrine

## Abstract

Aminomethylation reactions between chiral amino compounds (*S*)-(-)-1-phenylethylamine and l-proline with tetranonylresorcinarene and tetra-(4-hydroxyphenyl)resorcinarene in presence of formaldehyde were studied. The reaction between l-proline and resorcinarenes generated regioselectively chiral tetra-Mannich bases, due to the molecular incorporation of the fragment of the chiral amino acid. On the other hand, tetranonylresorcinarene and (*S*)-(-)-1-phenylethylamine formed regio- and diasteroselectively chiral tetrabenzoxazines, both by chiral auxiliary functionalization and by the transformation of the molecular structure that confers inherent chirality. The products obtained were characterized using IR, ^1^H-NMR, ^13^C-NMR, COSY, HMQC, and HMBC techniques. The reaction of (*S*)-(-)-1-phenylethylamine with tetra-(4-hydroxyphenyl)resorcinarene did not proceed under the experimental conditions. Once the chiral aminomethylated tetra-(4-hydroxyphenyl)resorcinarene was obtained, the chemical modification of poly(GMA–*co*–EDMA) was studied, and the results showed an efficient incorporation of the aminomethylated compound. For the physical modification, chiral aminomethylated tetranonylresorcinarenes were employed, finding that the incorporation of modified resorcinarenes occurs, but with less efficiency than that observed using chemical modification. The modified polymers were characterized via FT-IR, scanning electron microscopy imaging, and elemental analysis. Finally, polymers modified with chiral resorcinarenes were used as sorbents in norepinephrine microextraction; for practical purposes, artificial urine was prepared and used. To perform the microextraction, the decision was made to use the modern rotating-disk sorptive extraction technique (RDSE), because of its analytical attributes as a green, or eco-friendly, technique. According to the results, the method preliminarily validated for the determination of norepinephrine in artificial urine shows that the modified polymer with chiral derivative of tetra-(4-hydroxyphenyl)resorcinarene worked effectively as a new sorbent phase for the quantitative microextraction of norepinephrine, exhibiting high stability and homogeneity of composition and structure within the working range.

## 1. Introduction

L-norepinephrine (NE), L-noradrenaline (NA), L-arterenol, levarterenol or (*R*)-4-(2-amino-1-hydroxyethyl)-1,2-benzenediol (Figure 1) belong to the catecholamines (CA) group and are some of the most important monoamine neurotransmitters in the brain, because it is directly related to the central and autonomic nervous systems and has a strong influence on the regulation of the immune function. Changes in the concentration of NE are implicated in many common neuropsychiatric disorders [1,2].

The determination of NE levels in biological fluids is a useful tool for the diagnosis and monitoring of related diseases. For example, urinary catecholamine concentration values have been used as indicators of diseases such as pheochromocytoma and neuroblastoma. Therefore, more accurate measurements of CAs in biological samples are important both for clinical diagnosis and for the investigation of psychological conditions and psychosis [3].

Regarding the quantification of NE, the method widely used in clinical laboratories is HPLC-ECD (high performance liquid chromatography-electro chemical detection) [4,5], but due to noise and interference problems that arise, it is not suitable for routine analysis [2]. The use of HPLC-MS/MS (high performance liquid chromatography-tandem mass spectrometry) determination has taken off in recent years [6,7], but the high implementation costs and the requirement for specialized periodic maintenance make this method not very accessible in this field.

It is evident that the matrix of the sample makes the quantification of NE in biological samples difficult; for this reason, several sorbents have been used for the SPE technique to eliminate potential interference. The sorbents used in most procedures correspond to commercial cartridges for SPE, such as C18 [8,9], PBA (phenylboronic acid) [10,11], WCX (weak cation exchange resin) [12,13], and Oasis^TM^ HLB [11,12,14,15]. 

On the other hand, the techniques of microextraction in sorbent have enjoyed widespread analytical acceptance in recent years, mainly because they are miniaturized procedures or in general greener methodologies than conventional SPE. Sorbents have also been used for the extraction of norepinephrine by microextraction techniques such as: SPME (solid phase microextraction) using commercial fibers as sorbent [16,17] or developed by molecular imprinted polymers (MIP) [18], MEPS (microextraction by packed sorbent) using the commercial C18 stationary phase [19,20], and MSPE (magnetic solid phase extraction) using extractive phases based on nanoparticles [21,22].

In general, the implementation of these extractive and microextractive techniques for NE has markedly reduced the matrix effect, but the low selectivity of the sorbents used and the easy oxidation of NE have hampered routine analyses. A recent solution to this problem has been the development and application of sorbents with greater selectivity towards NE, for example using polymer surfaces modified with crown ether macrocycles [3,23,24].

Another similar alternative may be the synthesis and implementation of porous polymer matrices with immobilized chiral resorcinarenes, since there are several articles in the literature that describe stable complexes having been obtained in aqueous medium between NE and this type of macrocyclic structure [25,26]. 

The modification of materials for microextraction in sorbents with resorcinarenes is still in its infancy. There are only a few published articles that include the SPME technique. These studies include the synthesis and immobilization of various resorcinarenes of the tetraquinoxaline cavitand type and derivatives on commercial fibers for SPME, composed mainly of poly(dimethylsiloxane) (PDMS). With these modified fibers, it has been possible to efficiently and selectively preconcentrate analytes of environmental and pharmaceutical interest found at the trace level, such as aromatic hydrocarbons [27], safrole(5-(2-propenyl)-1,3-benzodioxole) and benzyl methyl ketone (BMK) [28], nitroaromatic explosives [29], benzene, toluene, ethylbenzene, and xylenes (BTEX) [30,31].

Resorcinarenes have been used as modifiers of polymeric materials both chemically and physically. The chemical modification involves the functionalization of the polymer by reaction with surface groups with the macrocycle, on either its lower or upper edge. On the other hand, in the physical modification, the macrocyclic system is fixed to the polymer by physisorption, where long hydrocarbon chains interact lipophilically. For this purpose, we have recently used methacrylate-based polymers, because they exhibit a high degree of porosity, surface area, and chemical stability, and low mechanical strength, in addition to the presence of highly reactive functional groups such as the epoxy group [32]. This type of modified polymer has shown high affinity with molecules of biological interest, such as peptides [33].

Continuing with our research on the synthesis and functionalization of resorcinarenes [34,35,36] and on the modification of polymeric materials [32,33], in this article we study the chemical and physical modification of methacrylate-based polymers with chiral resorcinarenes and their application as extractive phases for norepinephrine microextraction.

## 2. Materials and Methods 

### 2.1. General Experimental Information

The reagents and solvents were obtained from Merck (Darmstadt, Germany). A Nicolet^TM^ iS10 FT-IR spectrometer (Thermo Fisher Scientific, Waltham, MA, USA) with a Monolithic Diamond ATR accessory and absorption in cm^−1^ was used for recording the IR spectra. Raman spectra were recorded on a DXR Raman Microscope (Thermo Fisher Scientific, Waltham, MA, USA) with Raman shift in cm^−1^. The elemental analysis for carbon and hydrogen was carried out using a Flash 2000 elemental analyzer (Thermo Fisher Scientific, Waltham, MA, USA). Scanning electron microscopy (SEM) analysis was done on a VEGA3 SB microscope (TESCAN, Brno-Kohoutovice, Czech Republic). The samples were coated with gold in a 99.99% plasma state of purity in a metallizer. The thermogravimetric analysis was carried out with a thermogravimetric analyzer with large furnace (LF) TGA/DSC 3+ (Mettler Toledo, Columbus, OH, USA). 10 mg of each compound was heated from 30 °C to 1000 °C (heating rate of 5 °C/min). Nuclear magnetic resonance (^1^H NMR and ^13^C NMR) spectra were recorded on a BRUKER Avance 400 instrument (400.131 MHz for ^1^H and 100.263 MHz for ^13^C), and chemical shifts are given in δ units (ppm). HILIC-HPLC analyses were performed using an Agilent 1200 liquid chromatograph (Agilent, Omaha, NE, USA).

### 2.2. Preparation of Polymers

#### 2.2.1. Poly(GMA–co–EDMA) (**1**)

The procedure was adapted from the methodology of Okanda et al. [37]. The inhibitor of the monomers was removed via classical column chromatography, using silica gel 60 (0.063–0.200 mm particle diameter). A mixture of glycidyl methacrylate (GMA) (18%, 864 μL), ethylene dimethacrylate (EDMA) (12%, 570 μL), 1,1-azobis(cyclohexanecarbonitrile) (ABCN) (0.2%, 10 mg), cyclohexanol (58.8%, 3.1 mL) and dodecanol (11.2%, 0.56 g) was prepared. After homogenization, it was sonicated for 10 min at room temperature, and then nitrogen was bubbled through it for 10 min.

The mixture was transferred to an airtight container and heated to 57 °C for 40 h. The solid formed was cooled to room temperature, washed with EtOH (5 mL), and dried under vacuum at 55 °C until constant mass. Finally, the polymeric material was crushed and sieved at a particle size of 106 μm [38]. The chemical characterization was done using FT-IR, Raman spectroscopy, and the morphological characterization via TGA and SEM (see Appendix A). Elemental analysis: C = 62.05% and H = 7.75%.

#### 2.2.2. Poly(BuMA–co–EDMA) (**2**)

The same procedure was applied above, replacing the GMA monomer with butyl methacrylate (BuMA) (18%, 1.0 mL). The results of the characterization are shown in the Appendix A. Elemental analysis: C = 64.46% and H = 8.50%. 

### 2.3. Synthesis of Chiral Resorcinarenes

The preparation of starting resorcinarenes tetranonylresorcinarene (3) and tetra-(4-hydroxyphenyl)resorcinarene (**4**) was performed in accordance with previous studies [34,35]. The functionalization of **3** and **4** was developed based on previous research [33].

#### 2.3.1. Functionalization of Resorcinarene **3** with S-(-)-1-phenylethylamine

Formaldehyde 37% (2.5 mmol, 115 μL) was added to a solution of **3** (0.1 mmol) in a benzene-ethanol mixture (1:1) (9 mL) under constant stirring. Subsequently, a solution of *S*-(-)-1-phenylethylamine (0.5 mmol, 75 μL) containing NaOH 1 M (25 μL) in an EtOH/benzene mixture (1: 1) (1 mL) was added. The reaction was refluxed for 25 h under magnetic stirring. The mixture was then allowed to cool to room temperature, and the solvent was distilled to dryness. A solid was obtained, which was washed with EtOH (2 mL) and dried at room temperature for 24 h in the absence of light. The following was obtained:

15*H*,23*H*,26*H*,31*H*-6,32:8,14:16,22:24,30-Tetrametheno-2*H*,7*H*,10*H*,18*H*-cyclotetracosa[1,2-*e*:7,8-*e′*:13,14-*e″*:19,20-*e‴*]tetrakis[1,3]oxazine-5,13,21,29-tetrol,3,4,11,12,19,20,27,28-octahydro-3,11,19,27-tetrakis[(1*S*)-1-phenylethyl]-7,15,23,31-tetranonyl (**5**):

A faintly pink solid at a yield of 70%. Mp > 250 °C decomposition. IR (KBr/cm**^−^**^1^): 3347 (O-H), 1468 (C-O); ^1^H-NMR, CDCl_3_, *δ* (ppm): 0.89 (t, 12H, C-C-CH_3_), 1.28-1.37 (m, 68H, (CH_2_)_7_ and N-C-CH_3_), 2.17 (m, 8H, Ar-C-CH_2_), 3.73 (d, 4H, *J* = 16 Hz, ArCH-N), 3.80 (q, 4H, *J* = 8 Hz, NCH), 3.96 (d, 4H, *J* = 16 Hz, ArCH-N), 4.19 (t, 4H, *J* = 8 Hz, ArCH-Ar), 4.92 (d, 4H, *J* = 12 Hz, NCH-O), 5.13 (d, 4H, *J* = 12 Hz, NCH-O), 6.95 (t, 4H, *J* = 8 Hz, ArH *para* to CH-N), 7.04 (t, 8H, *J* = 8 Hz, ArH *meta* to CH-N), 7.11 (s, 4H, ArH *meta* to OH), 7.18 (d, 8H, *J* = 8 Hz, ArH *ortho* to CH-N), 7.66 (s, 4H, OH). ^13^C-NMR, *δ* (ppm): 14.1 (nonyl C-9), 21.4 (CH_3_-C-N), 22.7, 28.1, 29.4, 29.7, 29.8, 31.9, 32.0 (nonyl C-8, C-3, C-2, C-5, C-4, C-6, C-7), 32.7 (ArCHAr). 33.7 (nonyl C-1), 44.6 (ArCH_2_-N), 58.0 (ArCH-N), 80.9 (N-CH_2_-O), 108.9, 121.1, 123.5, 124.3 (benzoxazine C-5, C-8, C-9, C-7), 127.0, 127.1, 128.2, 144.5 (phenyl C-2, C-4, C-3, C-1), 148.7 (benzoxazine C-6, C-10). MALDI–TOF MS (4-nitroaniline) analysis showed a signal at *m/z* = 1597.35 corresponding to [M + Na]^+^ (calc. mass for M (C_104_H_140_N_4_O_8_): 1574.25). Anal. calcd. for (molecular formula, C_100_H_140_N_4_O_8_): C = 79.35, H = 8.96, and N = 3.56; found: C = 79.64, H = 8.84 and N = 3.61. 

#### 2.3.2. Functionalization of Resorcinarenes **3** and **4** with l-proline

NaOH 1 M (0.2 mmol) was added to a solution of 4 mL of l-proline (5 mmol) in distilled water. Then it was added to a solution of 60 mL of **3** (1 mmol) in ethanol-benzene mixture (1:1). Subsequently, formaldehyde 37% (12 mmol) was added dropwise. The mixture was brought to room temperature for 5 hours under constant stirring. A precipitate was formed that was washed with ethanol and dried in vacuum. The following was obtained:

2,8,14,20-Tetranonyl-5,11,17,23-tetrakis(*N*-(l-proline)methyl)pentacyclo[19.3.1.1^3;7^.1^9;13^.1^15;19^] octacosa-1(25),3,5,7(28),9,11,13(27),15,17,19(26),21,23-dodecaen-4,6,10,12, 16,18,22,24 octol (**6**): 

A creamy white solid at a yield of 75%. Mp > 250 °C (desc.). IR (KBr/cm**^−^**^1^): 3348 (O-H), 1669 (C=O); ^1^H-NMR, DMSO-*d*_6_, *δ* (ppm): 0.86 (t, 12H, CH_3_), 1.24-1.34 (m, 64H, (CH_2_)_8_), 1.68-1.83 (m, 8H, proline N-C-CH_2_), 2.22 (sb, 8H, proline N-C-CH_2_), 2.57 (m, 4H, proline N-C-CH_2_), 3.01 (m, 4H, proline N-CH_2_), 3.52 (t, 4H, proline CH), 3.93 (d, 4H, *J* = 16 Hz, ArCH-N), 4.02 (d, 4H, *J* = 16 Hz, ArCH-N), 4.24 (t, 4H, *J* = 8 Hz, ArCH-Ar), 4.31 (sb, 4H, ArO-H···N), 7.35 (s, 4H ArH). ^13^C-NMR, *δ* (ppm): 13.9, 22.1, 27.9, 28.7, 29.1, 29.2, 29.3, 29.3, 31.3 (nonyl C-1-C-9), 22.4, 32.5 (proline C-3 y C-2), 33.8 (ArCH-Ar), 49.0 (ArCH_2_-N), 53.1 (proline C-4), 66.0 (proline C-1), 107.4, 124.3, 124.4, 150.9 (resorcinol C-2, C-5, C-4, C1), 171.9 (C=O). MALDI–TOF MS (4-nitroaniline) analysis showed a signal at *m/z* = 1525.11 corresponding to [M + Na]^+^ (calc. mass for M (C_100_H_140_N_4_O_8_): 1502.01). Anal. calcd. for (molecular formula, C_88_H_132_N_4_O_16_): C = 70.30, H = 9.15 and N = 3.35; found: C = 70.55, H = 8.90 and N = 3.12 (because of its host nature, the compound occluding solvent molecules observed by NMR).

The reaction of l-proline with 4 was carried out following the procedure outlined by Maldonado et al. [33]. In this way, the chiral resorcinarene **7** was obtained:

2,8,14,20-Tetra-(4-hydroxyphenyl)-5,11,17,23-tetrakis(*N*–(l-proline)methyl)pentacyclo[19.3.1. 1^3;7^.1^9;13^.1^15;19^]octacosa-1(25),3,5,7(28),9,11,13(27),15,17,19(26),21,23-dodecaen-4,6,10,12,16,18,22,24-octol (**7**): 

An ocher yellow solid at a yield of 87%. Mp > 250 °C (desc.). IR (KBr/cm^−1^): 3413 (O–H), 1678 (C=O). ^1^H NMR, DMSO-*d*_6_, δ (ppm): 1.62–1.87 (m, 16H, proline NC–(CH_2_)_2_), 2.11–2.20 (m, 8H, proline NCH_2_), 3.88 (m, 4H, proline CH), 3.97 (m, 8H, NCH_2_Ar), 5.54–5.76 (m, 4H, ArCH), 6.20–6.55 (m, 20H, ArH). ^13^C NMR, δ (ppm): 22.8 and 23.4, 28.3 and 28.7, 41.9 and 42.2 (proline N(CH_2_)_3_), 49.7 and 49.9 (ArCH), 52.4 and 53.1 (ArCH_2_N), 67.1 and 67.4 (proline CH), 108.5 and 114.2 (resorcinol C–2), 121.8 and 122.9 (resorcinol C–5), 122.9 and 123.1 (hydroxyphenyl C–3), 129.6 and 129.7 (hydroxyphenyl C–2), 132.6 and 133.2 (hydroxyphenyl C–4), 150.9 and 151.9 (resorcinol C–4), 152.0 and 152.2 (hydroxyphenyl C–1), 154.8 and 154.9 (resorcinol C–1), 172.3 and 172.9 (C=O). Anal. calcd. for (molecular formula, C_76_H_76_N_4_O_20_): C = 61.95, H = 6.02 and N = 3.80; found: C = 61.30, H = 6.30 and N = 3.80 (because of its host nature, the compound occluding solvent molecules observed by NMR).

### 2.4. Polymeric Modifications of 1 and 2 with Chiral Resorcinarenes ***5***–***7***

The procedure was adapted from the methodology previously developed by our research group [32]. The chemical characterization was made via FT-IR, Raman spectroscopy, elemental analysis, and gravimetric analysis, and the morphological characterization via TGA and SEM. 

#### 2.4.1. Chemical Modification of Polymer **1** with Chiral Resorcinarene **7**

Chiral resorcinarene **7** (0.84 mmol) was dissolved in DMF/DMSO (8:2) (12 mL). Subsequently, the sieved polymer **1** (50 mg) was added, together with ground NaOH (0.84 mmol). The mixture was then stirred magnetically at 57 °C for 15 h. The mixture was cooled to room temperature, and the solid obtained was filtered under vacuum and washed with DMF until no residue of **7** was in evidence through monitoring via UV-Vis spectroscopy of the filtrate. The solid was then washed again with EtOH to neutral pH and dried at 57 °C for 2 h. Finally, the polymeric material was crushed and sieved to a particle size of 106 μm. In this way, the modified polymer **7**-poly(GMA–*co*–EDMA) (**8**) was obtained.

#### 2.4.2. Physical Modification of Polymer 2 with Chiral Resorcinarenes **5** and **6**

The same procedure as above was implemented, with the difference that it was carried out at room temperature for 24 h. In the case of chiral resorcinol 5, CHCl_3_ was used as the immobilization and washing solvent. In this way, the modified polymers 5-poly(BuMA–co–EDMA) (**9**) and 6-poly(BuMA–co–EDMA) (**10**) were obtained.

### 2.5. Microextraction and Quantification of Norepinephrine

The modified polymers **8**–**10** were applied as a sorbent phase, comparing the results with the unmodified polymers **1** and **2**. The modern technique of microextraction RDSE (rotary-disk sorption extraction) was implemented [39].

#### 2.5.1. RDSE Protocol

An aliquot of 10 mL of artificial urine [3,40] was taken, and it was doped at various concentrations with standard NE (in distilled and deionized water type 1 to pH 3) 1000 μgL^−1^ for preliminary tests and optimization, and 100–1000 μgL^−1^ for validation tests, vortexing it for 1 min. Then 20 mg of modified polymer was added to the rotating disk and conditioned with 5 mL of MeOH and with 5 mL of water at pH 3 for 15 min at 1200 rpm. Subsequently, the microextraction was carried out for 2 h at 1400 rpm. The elution was carried out with 9 mL of water at pH 3 for 41 min at 1200 rpm. The screening and optimization of the microextraction method was carried out using the Statgraphics Centurion XV software for Windows (Manugistics, Rockville, MD, USA).

#### 2.5.2. Detection and Quantification via HILIC-HPLC-UV

A 60-5-HILIC-D 4.6 × 150 mm (Kromasil) column was used, UV detector at 280 nm. The mobile phase was used in the isocratic mode and consisted of distilled and deionized water type 1 (0.05% TFA)/MeCN (0.05% TFA) (95:5), flow of 1.0 mLmin^−1^, injection volume of 100 μL, room temperature of the oven and analysis time 5 min.

## 3. Results

### 3.1. Preparation of Polymers

Polymers poly(GMA–*co*–EDMA) (**1**) and poly(BuMA–*co*–EDMA) (**2**) were synthesized following a procedure that was adapted from the methodology of Okanda et al. [37]. As depicted in Scheme 1, EDMA and BuMA (or GMA) were mixed in cyclohexanol:dodecanol mixture and reacted at 57 °C in presence of 1,1-azobis(cyclohexanecarbonitrile) (ABCN) to form **1** and **2**, respectively. The chemical characterization was performed via FT-IR, Raman spectroscopy, and elemental analysis and the morphological characterization via TGA and SEM. The polymers had been previously synthesized by other authors [41,42], and our experimental data agreed with those reported.

### 3.2. Synthesis of Chiral Resorcinarenes

For the present study, we chose the resorcinarenes tetranonylresorcinarene (**3**) and tetra(4-hydroxyphenyl)resorcinarene (**4**) for the aminomethylation reaction with chiral aminocompounds (*S*)-(-)-1-phenylethylamine and l-proline. The synthesis of resorcinarenes **3** and **4** was carried out starting from resorcinol with the respective aldehyde in a mixture of ethyl alcohol and water (1:1) at reflux, in accordance with how it is described in the literature and as we have previously developed it [34,35]. Once the resorcinarene tetranonylresorcinarene (**3**) was obtained, the first step involved reaction with the chiral amine (*S*)-(-)-1-phenylethylamine in presence of formaldehyde via base-catalyzed cyclocondensation, as per the method described in the literature (Scheme 2) [43]. The product obtained in the reaction was easily purified by means of recrystallization and was characterized via spectral techniques, including FT-IR, ^1^H-NMR, ^13^C-NMR, and 2D-NMR experiments (see Appendix A). In this way, the aminomethylation product of **3** with (*S*)-(-)-1-phenylethylamine showed that FT-IR analysis was in agreement with the organic functionalities present, the hydroxyl group stretches at 3347 cm^−1^ (O-H) and 1468 cm^−1^ (C-O) being the principal features observed, whereas the C-N of the amine group can be seen at 1286 cm^−1^; the bands of the alkyl substituent and the aromatic ring can also be seen at 2923 and 3027 cm^−1^, respectively.

The ^1^H-NMR spectrum of **5** displayed characteristic signals corresponding to the nonyl substituent at 0.89 ppm for methyl groups and in the range of 1.28–2.17 ppm for the aliphatic chain. The methine bridge was observed at 4.19 ppm. Protons of the ethyl-benzyl amine fragment were displayed at 1.37 and 3.80 ppm and aromatic protons at 6.95, 7.04, and 7.18 ppm. Also, in the aromatic region, a singlet signal for *meta*-protons of resorcinarene moiety was observed at 7.11 ppm. The signal for the hydroxyl groups was observed at 7.66 ppm, but it only integrated for four protons. In this sense, the disappearance of one of the hydroxyl signals is indicative of the formation of a benzoxazine ring, which was confirmed with two groups of signals: first, the signals for diastereotopic protons Ar-CH_2_-N at 3.76 and 3.96 ppm, the two signals with doublet multiplicity and a coupling constant of *^2^J* = 16 Hz, and second, the signals for diastereotopic protons O-CH_2_-N at 4.92 and 5.13 ppm, the two signals with doublet multiplicity and a coupling constant of *^2^J* = 12 Hz. As seen in the Appendix A, the high degree of symmetry observed in the signals indicates the formation of a single isomer, as will be discussed later. In order to confirm the above assignments, the ^1^H-^1^H COSY spectrum was recorded. The correlations for **5** are given in Table 1; the observed ^1^H-^1^H COSY correlations confirm the connectivity and the assignments made. 

Carbon signals in the ^13^C-NMR were unambiguously assigned through 1D- and 2D-NMR experiments, including the HMQC and HMBC spectra. The observed correlations are given in Table 2. ^13^C-NMR spectrum shows a weak signal at 144.5 ppm with no correlation in the HMQC spectrum. In the HMBC spectrum, this signal shows a correlation with the methyl protons at 1.37 ppm and with the signal at 3.80 ppm; hence these signals must be due to an ethyl-benzylic amine fragment. There are six weak signals at 108.9, 123.5, 124.3, 148.7, and 149.6 ppm. These signals have no correlation with the HMQC spectrum and were attributed to the resorcinarene fragment.

In the HMBC spectrum, the signal at 108.9 ppm exhibits a correlation with the diastereotopic methylene group protons at 3.73 and 3.96 ppm, and the signal at 149.6 ppm exhibits a correlation with the diastereotopic methylene group protons at 3.73, 3.96, 4.92, and 5.13 ppm. The signal at 44.6 ppm exhibits a correlation with the diastereotopic methylene group protons at 3.73 and 3.96, and the signal at 80.9 ppm exhibits a correlation with the diastereotopic methylene group protons at 4.92 and 5.13 ppm in HMQC spectrum, so these signals were attributed to the benzoxazine ring. The signal at 123.5 ppm has a correlation with the signal at 4.19, so the signal is probably due to the aromatic carbon in the resorcinarene ring attached to bridged methine (Table 2). 

The aminomethylation between **3** and l-proline was done by direct reaction with formaldehyde solution (Scheme 2). In this way, derivative 6 was obtained as a creamy white solid, and the molecular weight was determined via MALDI-TOF/MS (*m/z* = 1525.11 corresponding to [M + Na]^+^). The FT-IR spectrum of 6 showed amine group (1044 cm^−1^ and 773 cm^−1^), carbonyl group (1669 cm^−1^), aromatic ring (1609 cm^−1^), alkyl chain (2929 cm^−1^), and hydroxyl group (3348 cm^−1^) absorptions. The ^1^H-NMR spectrum displayed characteristic signals for the resorcinarene system: nonyl chains (0.86, 1.24, to 1.34 ppm), a methylene bridge fragment between the aromatic rings (4.24 ppm), and the aromatic hydrogen of a pentasubstituted resorcinol unit (7.34 ppm). The strong cyclic hydrogen bonding in the product is the response of the signals for diastereotopic protons Ar-CH_2_-N at 3.93 and 4.02 ppm, the two signals with doublet multiplicity and a coupling constant of *^2^J* = 16 Hz. The signals for the proline fragment are at 1.68–1.83, 2.22, 2.57, 3.01, and 3.52 ppm, and they are consistent with the chemical shifts of pure proline. The proton involved in the hydrogen bonding O-H∙∙∙N appeared at 4.31 ppm, like a wide singlet. As seen in the Appendix A, this behavior is clear evidence of diasterotopic protons, which proves the strength of the possibly intramolecular hydrogen-bond interaction O-H∙∙∙N, granting the molecule greater structural rigidity due to the formation of a stable cycle of 6 members. As seen in Figure 2, the signal at 4.31 ppm is wide and hides some of the signals of the diasterotopic hydrogens. The addition of D_2_O allowed observing the signals of this hydrogen, confirming that the stability of the intramolecular hydrogen bonding confers rigidity to the system and that the reaction is regioselective towards the formation of a single stereoisomer. The carbon signals in the ^13^C-NMR spectra were unambiguously assigned through 1D- and 2D-NMR experiments, including HMQC and HMBC. 

The reaction between tetra(4-hydroxyphenyl)resorcinarene (**4**) and (*S*)-(−)-1-phenylethylamine generated a complex mixture difficult to solve, according to that observed in the RP-HPLC analysis; however the reaction between **4** and l-proline passed efficiently as we previously published with the same spectroscopy results [33].

The chirality characteristic of this type of derivative is solved by performing an analysis of the results achieved by NMR analysis. First, the high degree of symmetry of the signals and the small degree of complexity exhibited in the ^1^H-NMR spectra of products shows that the reaction is regioselective. This condition is also fulfilled for the functionalized resorcinarenes 6 and 7, where the structural rigidity is acquired by the formation of the four intramolecular hydrogen bridges of type ArO-H∙∙∙N. The high selectivity (specificity) of this type of reaction can be attributed to the stereochemistry of the chiral amino compound, since the presence of a stereogenic center directly linked to the amino group promotes the formation of a single epimer, presumably via a "gearing" mechanism [44]. In this way, the chirality of the **5**–**7** derivatives is provided by three structural elements: The coupling of four chiral amino compound entities to the macrocycle, the formation of four stereogenic centers in the methine bridges that were initially prochiral in the starting resorcinarene, and the lack of symmetry in the *crown* of the three-dimensional macrocyclic structure.

### 3.3. Polymeric Modifications of **1** and **2** with Chiral Resorcinarenes ***5***–***7***

The modification of the polymers was carried out by two routes: for polymer **1**, the chemical modification was done by reaction with the exposed epoxide ring and the hydroxyl group on the lower rim of resorcinarene 7, and in second route the physical modification of polymer 2 was carried out with the aminomethylated resorcinarenes 5 and 6.

#### 3.3.1. Chemical Modification of Polymer **1** with Chiral Resorcinarene **7**

For the present research, we performed the fixation of resorcinarene **7** by direct reaction with the polymer’s glycidyl group in basic media with the hydroxyl group in the lower rim of the chiral resorcinarene **7** (Scheme 3). Modified polymer **7**-poly(GMA-*co*-EDMA) (**8**) was purified and then characterized using ATR-FTIR and Raman spectroscopy in the dry state at room temperature. The ATR-FTIR spectra of the starting polymer 1 and modified polymer 8 are shown in Figure 3a. The main difference observed in the ATR-FTIR spectra is the presence of broad and intense absorption at 3377 cm^−1^ corresponding to the O–H groups of resorcinarene rings and the disappearance of the epoxy group C-O band at 1083 cm^−1^.

To confirm these results, the sample was analyzed via Raman spectroscopy (Figure 3b), which shows different signals compared to the unmodified material; specifically, the absorption at 3090 cm^−1^ is present. This signal is characteristic of aromatic systems and confirms that the chiral resorcinarene 7 was fixed at polymer 1. Comparative thermogravimetric analysis (TGA) was performed at a heating rate of 10 °C/min under nitrogen. The thermal decomposition temperatures and half-decomposition temperatures of polymer 1 and modified polymer **8** indicated that the thermal stability of the modified polymer is slightly greater than that of the unmodified polymer. The derivative of the TGA thermogram (Figure 3c, blue line) mainly reveals a loss of mass, attributed to the decomposition of organic matter at 328 °C. So, the modified polymer **8** is thermostable below 230 °C for short periods of time, presenting a TGA profile similar to the starting polymer **1**. 

The results of SEM at 2 and 5 μm (Figure 3d,e) showed that the morphology of **8** exhibits a pattern analogous to **1** in terms of porosity, uniformity and microglobular structure; these results are equivalent to other investigations of chemical modification of poly(GMA-*co*-EDMA) [45].

The amount of fixed chiral resorcinarene 7 was determined from the information of the nitrogen percentage in the elemental analysis. These results show that the polymer poly(EDMA-*co*-GMA) has a high degree of substitution of **7** on the surface. This result was also confirmed by gravimetric analysis, and it was established that the degree of substitution was 413 μmol/g.

#### 3.3.2. Physical Modification of Polymer 2 with Chiral Resorcinarenes **5** and **6**

The physical modification of polymer **2** by "immobilization by adsorption" with **5** and **6** was carried out by applying the methodology described in the experimental section, without the addition of NaOH and using CHCl_3_ and DMF: DMSO (8: 2), respectively, as solvent. In this way, it was possible to obtain the modified polymers **5**-poly(BuMA-*co*-EDMA) (**9**) and **6**-poly(BuMA-*co*-EDMA) (**10**) (Scheme 4). Each macrocyclic structure was immobilized on polymer **2**, mainly due to lipophilic interactions between the nonyl hydrocarbon chains of the functionalized resorcinarene and the butyl hydrocarbon chains of the polymer surface.

Modified polymers **9** and **10** were chemically characterized via ATR-FT-IR, Raman, elemental analysis, and gravimetric analysis and morphologically via TGA and SEM, as in the chemical modification experiment, and the results are shown in the Appendix A.

When comparing the physical fixation of chiral resorcinarenes with respect to chemical fixation, it was found that it is less efficient, since with **9** and **10** there is a fixation of 102 and 116 μmol/g, respectively.

### 3.4. Microextraction and Quantification of Norepinephrine

Modified polymers **8**–**10** were applied as sorbent phases, contrasting the results with base polymers 1 and 2. Human urine was used as a matrix, and for practical purposes artificial urine was prepared and used, as has been done in recent studies for the same purpose [3,23,40]. To perform the microextraction, it was decided to use the modern rotary disk sorption extraction technique (RDSE), because of its analytical attributes as a green, or eco-friendly, technique [39,46,47,48].

The first preliminary trial consisted of establishing which polymer allows the greatest recovery of norepinephrine, in terms of the analytical signal (area). The RDSE protocol was applied in triplicate, doping the artificial urine at 1000 μgL^−1^ and using 5 mL of water for 5 min in the elution stage. Figure 4 shows that the use of **8** as the sorbent phase exhibited the highest analytical norepinephrine signal, and also significantly higher than the other polymers. Therefore, subsequent microextraction tests were performed using only polymer **8**.

For screening, a two-level factorial design (2*^K^*) with 4 centers (*K* = 4) was used for a total of 20 experiments. The experimental factors were: sorbent phase mass (20 and 30 mg), desorption time (20 and 50 min), millimolar concentration of the eluent (0 and 40 mM), and volume of eluent (5 and 10 mL). The Pareto chart standardized for recovery is shown in Figure 5, where the standardized effect bars show that only the factors of eluent volume and desorption time have an effect on the response (recovery). Sorbent phase mass, concentration of the eluent and the interactions (AB, AC, AD, BC, BD, and CD) between the four variables showed no statistically significant effect. 

Subsequently, the factors eluent volume and desorption time were optimized, applying a central composite design (2*^K^* + 2*K* + C), centered on the faces and with two centers, for a total of 10 experiments [49]. The levels of each factor were volume of eluent: 5.0, 7.5 and 10 mL, and desorption time: 15, 30, and 45 min. Figure 6 shows the Pareto chart, where the standardized effect bars reveal that both the volume of eluent and the time of desorption have a marked influence on the recovery of the analyte. The factor with the greatest effect is the volume of eluent, and the interactions between the two factors have a very weak effect on the response.

Optimal conditions for a maximum recovery value (~100%) were obtained with the response surface graph, which shows the combination of factorial levels (Figure 7). According to that observed in the contour plot, using a desorption time between 33 (0.2) and 45 min (1.0) and a volume of eluent between 9 (0.4) and 10 mL (1.0), a recovery of norepinephrine greater than 95% would be achieved, which satisfies the requirements of the analytical procedure. The optimized response to the maximum value reveals that it is possible to obtain a recovery close to 99%, according to the following coded and natural values (punctual): desorption time 0.661775 (41 min) and volume of eluent 0.705685 (9 mL). In this way, the norepinephrine microextraction method was successfully optimized, using the new modified polymer **8** as a sorbent phase, with a concentration factor of 10 (FC = 10).

Finally, most validation tests of the method were carried out according to the criteria suggested by FDA and ICH [50]. The linearity of the RDSE method was performed at six concentration levels and showed optimal linearity (R^2^ > 0.99). The parameters of accuracy (bias) and precision (repeatability) are presented in terms of recovery (%) and RDS (%), respectively (Table 1). These values were calculated with six independent enriched samples for each concentration level, taking into account the extreme values of the validated interval (50 and 500 μgL^−1^). The accuracy results of 96.2% and 101.7% and precision of 7.0% and 5.6% are statistically acceptable. The limit of detection (LOD) and the lowest limit of quantification (LLOQ) were estimated based on the signal-to-noise ratio, using the 3-σ and 10-σ approach, respectively, using eight extracts enriched at 50 μgL^−1^ (Table 3). The values obtained are sufficiently low (11.3 and 34.0 μgL^−1^) for the determination of norepinephrine in clinical analysis.

According to the results of Table 1, the method preliminarily validated for the determination of norepinephrine in artificial urine allows obtaining quantitative recoveries throughout the linear range. These results show that the modified polymer **8** worked perfectly as a new sorbent phase for the quantitative microextraction of norepinephrine, exhibiting high stability and homogeneity of composition and structure within the working range.

Additionally, the limits of quantification LLOQ and ULOQ show the abnormal urinary levels of the biochemical marker norepinephrine, which are generally higher than the normal levels (~10–30 μgL^−1^) [19]. For example, patients with unbalanced norepinephrine secretions present approximate levels of this neurotransmitter in diseases such as pheochromocytoma (~89 μgL^−1^), drug addiction (~65 μgL^−1^), and depression (~47 μgL^−1^), among others [19,51].

Table 4 shows a comparison of the analytical characteristics of the sample preparation method developed with respect to other recently published articles. Among the extractive techniques, RDSE turns out to be the simplest and most economical, due to the fact that it employs as a device a low-cost, easily accessible and 100% reusable rotatory disk, compared to the high costs of commercial devices for SPE and MEPS and of the specific reagents needed for DLLME and CM-LPME-SSP. On the other hand, the detection system used in the present method (UV), is also the one with the lowest cost, which significantly reduces costs in routine laboratories. In addition, the mobile phase used in the quantification is one of the simplest, in contrast to other methods that apply organic solvents, organic and inorganic acids, strong bases, surfactants, chiral selectors, and inorganic salts.

This makes the composition of the mobile phase more complex, exponentially increasing the operating costs and the optimization time of the chromatographic method. Additionally, the use of solid reagents in the mobile phase can lead to partial or total obstruction of the chromatographic lines, significantly affecting the detection results. Finally, the present method of sample preparation provides recovery, precision, LOD, and linear range values similar to other investigations. This reflects the versatility of the RDSE technique for use in different matrices, characterized by being a microextractable and eco-friendly methodology without losing the quality and rigor of the analytical method. 

## 4. Conclusions

The reaction of the resorcinarenes **3** and **4** with l-proline generated regioselectivity chiral tetra-Mannich bases **6** and **7**, whereas the reaction between (*S*)-(-)-1-phenylethylamine and tetra-(4-hydroxyphenyl)resorcinarene (**4**) formed both regio- and diastereoselectivity chiral tetrabenzoxazine **5**. Aminomethylation reactions were efficiently carried out with percentage yields between 70% and 87%. The high selectivity of this type of reaction can be attributed to the stereochemistry of the chiral amino compound, since the presence of a stereogenic center directly linked to the amino group promotes the formation of a single epimer. Chemical modification of poly(GMA–*co*–EDMA) showed an efficient incorporation of the aminomethylated compound **7** while for the physical modification of poly(BuMA–*co*–EDMA) with chiral aminomethylated tetranonylresorcinarene **5** and **6** occurs but with less efficiency. The microextraction of norepinephrine in artificial urine via the rotating-disk sorptive extraction technique (RDSE) using modified polymer shows that the modified polymer with chiral derivative of tetra-(4-hydroxyphenyl)resorcinarene worked perfectly as a new sorbent phase for the quantitative microextraction of norepinephrine (**7**), exhibiting high stability and homogeneity of composition and structure within the working range. Finally, the present method of sample preparation provides recovery, precision, LOD, and linear range values similar to other investigations.

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
