# Peer review of "Preparation of Methacrylate-Based Polymers Modified with Chiral Resorcinarenes and Their Evaluation as Sorbents in Norepinephrine Microextraction"

_polymers, 2019, doi:10.3390/polym11091428_

Round 1

Reviewer 1 Report

The manuscript entitled “Preparation of methacrylate-based polymers modified with chiral resorcinarenes and their evaluation as sorbents in norepinephrine microextraction” by Castillo-Aguirre and Maldonado describes the investigation of aminomethylation reactions between chiral amino compounds (namely (S)-(-)-1-phenylethylamine and L-proline) with resorcinarenes  (tetranonylresorcinarene  and  tetra-(4-hydroxyphenyl)resorcinarene )) in the presence of formaldehyde.  When the authors reacted L-proline with the resorcinarenes yielded regioselectively chiral tetra-Mannich bases, which were obtained via the incorporation of the fragment of the chiral amino acid.  In contrast, the reaction between tetranonylresorcinarene and (S)-(-)-1-phenylethylamine yielded  regio- and diasteroselectively chiral tetrabenzoxazines, which were obtained via chiral auxiliary functionalization and via the transformation of the molecular structure that had conferred inherent chirality.  The authors found that the reaction between (S)-(-)-1-phenylethylamine with tetra-(4-hydroxyphenyl)resorcinarene did not proceed under the conditions employed for this study.

            When the chiral aminomethylated tetra-(4-hydroxyphenyl)resorcinarene was obtained, it was used to covalently functionalize or chemically modify the random copolymer poly(GMA–co–EDMA), and it was found that the chiral resorcinarene was successfully incorporated into this copolymer.  In addition, physical modification was also investigated by the authors with the use of chiral aminomethylated tetranonylresorcinarenes.  Although the physical modification was also successful, they found that this modification was less efficient than the chemical modification.  The potential use of the resorcinarene-modified polymers as sorbents for the extraction of norepinephrine (NE) was investigated, with artificial urine serving as a matrix from which the extraction was performed.  These microextraction experiments were performed via the rotating disk sorptive-extraction (RDSE) technique.

            The authors have employed a variety of characterization techniques, including 1H and 13C NMR spectroscopy, as well as 2D-NMR techniques such as COSY, HMQC and HMBC.  Other techniques include Raman and FT-IR spectroscopy, as well as elemental analysis, scanning electron microscopy (SEM), thermogravimetric analysis (TGA), and HILIC-HPLC.  Overall, the results are very well-supported by the characterization methods employed in this manuscript.  Perhaps a soft ionization mass spectroscopic analysis (such as possibly ESI-MS or MADI-MS) would also be of interest to complement these results, but I believe that the results provided by the authors is sufficient (also possibly the mass spectroscopic techniques may encounter problems due to the molecular weight of the polymers).

            The modified polymers reported herein have great potential for use as sorbents for quantitative microextraction of NE, and limit of detection (LOD) as well as lowest limit of quantification (LLOQ) values suitable for clinical analysis were obtained.  The material resented in this manuscript will be of interest to researchers in a variety of fields, including polymer science, supramolecular chemistry, analytical chemistry, as well as those in the biomedical fields.  This work fits well within the scope of Polymers, and is of interest both from a fundamental scientific viewpoint as well as from a practical aspect.  The sorbent materials presented in this work have significant potential for application in the diagnosis of diseases.  My recommendation is for this manuscript to be accepted for publication, with minor revisions, with suggestions for possible revisions provided in further detail below.

            The presentation of this manuscript is excellent.  It is written, is very clear, and is well-organized.  I have only some minor suggestions below:

            Although the abbreviated names of the random copolymers Poly(GMA–co–EDMA) and Poly(BuMA–co–EDMA)  are provided in this manuscript, it may be recommended for a full name to be given at some point in this manuscript (not for regular use) but simply for definition purposes when the abbreviated names are first mentioned. 

Page 1, abstract, last sentence: “worked perfectly as a new” can possibly be changed to “

 worked effectively as a new”.

The indentation of the paragraphs seems to be inconsistent.  The beginnings of some paragraphs are indented, but others are not.  The authors may need to check with the MDPI guidelines to check that the formatting matches that employed by Polymers.

Page 1, introduction, lines 2-3: “belongs to the catecholamines (CA) group and is one of the most important” can possibly be changed to “belong to the catecholamines (CA) group and are some of the most important”.

Page 2, 2nd to the last paragraph, line 1: “modification of materials for microextraction in sorbent” can possibly be changed to “modification of materials for microextraction in sorbents”.

Page 3, section 2.2.1, line 5: “After homogenizing, it” can possibly be changed to “After homogenization, it”.

Page 3, section 2.2.1, second paragraph, line 4: “was done us” can be changed to “was done using” or “was performed using”.

Page 4, section 2.3.2 title: “3 y 4 with” can possibly be changed to “3 and 4 with”.

Page 4, section 2.3.2, line 2: “Then it is added” can be changed to “Then it was added”.

Page 4, section 2.4.1, last line: “7-poly(GMA–co–EDMA)” can be changed to “7-poly(GMA–co–EDMA)” (with “co” in italics font).

Page 4, section 2.4.1:  The section number 2.4.1 appears twice. The second section with this number, “2.4.1. Physical Modification of Polymer 2 with Chiral Resorcinarenes 5 and 6“ can be renumbered as section 2.4.2 to give “2.4.s. Physical Modification of Polymer 2 with Chiral Resorcinarenes 5 and 6 “. 

Page 5, Section 2.5.1, line 1: “was taken, doping it at various concentrations with” can possibly be changed to “was taken, and it was doped at various concentrations with”.

Page 6, Section 2.5.2, line 2: “in isocratic mode” can be changed to “in the isocratic mode”.

Page 7, line 1: “signals of nonyl substituent at 0.89” can possibly be changed to “signals corresponding to the nonyl substituent at 0.89”.

Page 7, 2nd paragraph, line 1: “signals in 13C-NMR” can possibly be changed to “signals in the 13C-NMR spectra”

Page 8, 1st paragraph, line 7: “due to aromatic carbon” can possibly be changed to “due to the aromatic carbon” or possibly; “due to an aromatic carbon”.

Page 8, 2nd paragraph, line 20: “in 13C-NMR were” can possibly be changed to “in the 13C-NMR spectra were”.

Page 10, Scheme 3 caption: “poly(GMA-co-EDMA)” can be changed to “poly(GMA-co-EDMA)” (with “co” changed to italics font).

Page 11, line 2: “specifically, absorption at” can possibly be changed to “specifically, the absorption at”.

Page 12, Scheme 4 caption: “poly(GMA-co-EDMA)” can be changed to “poly(GMA-co-EDMA)” (with “co” changed to italics font).

Page 12, section 3.4, line 2: “used as matrix” can be changed to “used as a matrix”.

Page 12, section 3.4, second paragraph, line 1: “consisted in establishing” can be changed to “consisted of establishing”.

Page 13, Figure 5: Possibly error bars can be included for the Pareto chart in Figure 5.

Page 14, Figure 6: Possibly error bars can be included for the Pareto chart in Figure 6.

Page 16, Table 4:  Possibly error margins can be added to the values reported in Table 4.

Author Response

Please see the attachament

Reviewer 2 Report

Authors described the preparation of methacrylate-based polymer networks that are capable of micro extraction of norepinephrine after chemical or physical modification. The work includes the precise structural investigation and detailed synthetic procedure. Application also would be interesting to some audience. However, some supplementary data should be necessary, which gives scientific soundness and makes the story stronger and more appealing.

Some technical issues are follows:

please detail or compare the effects from proline or 1-phenylethylamine when absorbing norepinephrine. It would help readers to understand the working principle of polymers that authors designed. Absorption capability of non-modified polymers should be given and compared with modified ones. Absorption capabilities of 9 and 10 should be shown although they were found to be very low. The x axis of the graph in figure 4 did not match with the main text. Explanation for y axis in figure 5 or 6 should be given. For example, what are BB AB AA for? why are they important? What happens if 5 has hydroxyl groups that induce chemical modification? Can it show better performance?

Round 2

Reviewer 2 Report

Authors answered well overall. I found that some parts still need to be revised for clarification in terms of scientific clarity and readability, but I believe it would be improved in future. I recommend the manuscript for publication in its current form.